# A potential marker of radiation based on 16S rDNA in the rat model: Intestinal flora

**Liying Zhang[1‡], Zhiming Miao[1‡], Yangyang Li[1], Xiaomin Xu[1,2], Ting Zhou[1], Yiming Zhang[1], Yongqi Liu** [1,3]*

1 Provincial-Level Key Laboratory for Molecular Medicine of Major Diseases and The Prevention and Treatment with Traditional Chinese Medicine Research in Gansu Colleges and Universities, Gansu University of Chinese Medicine, Lanzhou, Gansu, China, 2 Medical College of Hexi University, Zhangye, Gansu, China, 3 Key Laboratory of Dunhuang Medicine and Transformation at Provincial and Ministerial Level, Gansu University of Chinese Medicine, Lanzhou, Gansu, China

‡ LZ and ZM are co-first authors on this work.

* liuyongqi73@163.com

**Data Availability Statement:** All relevant data are within the manuscript and its Supporting Information files.

**Funding:** We confirm that the following projects have funded us National Natural Science

## Abstract

The gastrointestinal microbiota plays an important role in the function of the host intestine. However, little is currently known about the effects of irradiation on the microorganisms colonizing the mucosal surfaces of the gastrointestinal tract. The aim of this study was to investigate the effects of X-ray irradiation on the compositions of the large intestinal Microbiotas of the rat. The gut microbiotas in control mice and mice receiving irradiation with different dose treatment were characterized by high-throughput sequencing of the bacterial 16S rDNA gene and their metabolites were detected by gas chromatography-mass spectrometry. Unexpectedly, the diversity was increased mildly at 2Gy irradiation, and dose dependent decreased at 4Gy, 6Gy, 8Gy irradiation. The phyla with large changes in phylum level are Firmicutes, Bacteroides and Proteobacteria; the abundance ratio of Firmicutes/Bacteroides is inverted; and when 8Gy is irradiated, the phylum abundance level was significantly increased. At the genus level, the abundance levels of *Phascolarctobacterium*, *Ruminococcaceae* and *Lachnospiraceae* increased at 2Gy irradiation, and significantly decreased at 4Gy, 6Gy, and 8Gy irradiation; the abundance level of *Prevotellaceae* diminished at 2Gy irradiation, and enhanced at 4Gy, 6Gy, 8Gy irradiation; The abundance level of *Violet bacteria* (*Christenellaceae*) and *Lactobacillus* attenuated in a dose-dependent manner; *Lachnoclostridium* enhanced in a dose-dependent manner; *Bacteroides* was in 4Gy, 6Gy, 8Gy The abundance level increased significantly during irradiation; the abundance level of *Shigella* (*Escherichia-Shigella*) only increased significantly during 8Gy irradiation. Lefse predicts that the biomarker at 0Gy group is *Veillonellaceae*, the biomarker at 2Gy group is *Firmicutes*, the biomarkers at 4Gy group are *Dehalobacterium* and *Dehalobacteriaceae*, the biomarkers at 6Gy group are *Odoribacter*, and the biomarkers at 8Gy group are *Anaerotruncus*, *Holdemania*, *Proteus*, *Bilophila*, *Desufovibrionales and Deltaproteobacteria*. Overall, the data presented here reveal that X-ray irradiation can cause imbalance of the intestinal flora in rats; different doses of irradiation can cause different types of bacteria change. Representative bacteria can be selected as biomarkers for radiation damage and repair.This may contribute to the development of radiation resistance in the future.

Foundation of China, 82260882 Li-Ying Zhang National Natural Science Foundation of China, 82004094 YongQi Liu Longyuan Youth Innovation and Entrepreneurship Talent Project in 2021, GZTZ [2021]17-1 Li-Ying Zhang Natural Science Foundation of Gansu Province, 20JR10RA318 Li-Ying Zhang Natural Science Foundation of Gansu Province, 20JR10RA332 YongQi Liu China Postdoctoral Science Foundation Project, 2021M693794 Li-Ying Zhang Lanzhou City Health Key Science and Technology Development Project, 2021006 Li-Ying Zhang Funders YongQi Liu and Li-Ying Zhang conceived and designed the experiments, and provided the funding required for the experiments.

**Competing interests:** NO authors have competing interests Enter: The authors have declared that no competing interests exist.

## Introduction

Many studies have highlighted the presence of as many as 100 trillion bacterial species in the human gut, which represent several times as many cell populations as somatic and germ cells in the human host [1–3]. These commensal bacteria of the gut are collectively called "gut microbiota," and increasing evidence suggests that the composition of gut microbiota is closely related to the health of the host, including the onset of disease [4, 5]. Increasing evidence shows that the human gut microbiome plays a role in keeping the organism in a steady state, and intestinal dysbacteriosis is often accompanied by obesity [6], cardiovascular disease, inflammation disease [7], immune function disorder [8], metabolic disease [9–11] and even cancer [12]. Therefore, homeostasis of gut microbiota plays an important role in maintaining human health.

X-ray is one of the most used radiation therapies in clinic. Casero found after irradiation, it can cause DNA double strand or single strand breaks and other biological effects. David C et al in 2017 found it has been shown that the gut microbiota regulates the efficacy of radiotherapy and chemotherapy through a "TIMER" mechanism, which suggests a reduction in translocation, immune regulation, metabolism, enzymatic degradation, and diversity. Gut microbiota [13] and most tissue cells [14, 15] in the body are sensitive to X-ray, especially those with active proliferation. Casero found that the mouse gut microbiome was response to $^{16}O$ radiation, dose-dependent changes in microflora diversity and composition of specific groups [16]. Zhang also shows that gut microbiome is very sensitive to IR and can be used as radiation biomarkers [17]. Recent reports suggest that metabolomics has shown increased concentrations of microbially derived propionic acid and tryptophan metabolites in feces of elite survivors, and use of these metabolites can lead to long-term radiation protection, relieve hematopoietic and gastrointestinal syndromes, and reduce pro-inflammatory responses [18]. However, there are few reports about the effects of different doses of X-ray radiation on intestinal flora.

In this report, different doses of X-ray irradiation were used to detect the levels of intestinal flora and metabolites in feces of rats to observe the effects of different doses of X-ray irradiation on intestinal flora in SD rats.

## Materials and methods

### Radiation

X-RAD-225 X-ray source (Precision, USA) was used to generate X-ray, the dose rate of which was about 200cGy/min (225.0 kV, 13.3 mA). The absorbed dose was 0Gy, 2Gy, 4Gy, 6Gy, 8Gy in each group. All irradiations were carried out at room temperature.

### Animals and groups

Fifty SD rats (6 weeks of age to 8 weeks of age) were purchased from Lanzhou University. The animal certificate number was 62000800000176 and animal license number was SCXK (Gan) 2013–0002. All animal experiments reported in this study were approved by the Ethics Committee of Gansu University of Chinese Medicine. The environment temperature was 22±2°C, the relative humidity was 55±10%, and the environment alternates light and dark were 12 hours.

Fifty SD rats were randomly assigned into 0Gy, 2Gy, 4Gy, 6Gy, 8Gy groups ($n$ = 10 in each group). Rats in each group were treated with 0Gy, 2Gy, 4Gy, 6Gy, 8Gy dosses of X-ray respectively each time and the rats were sacrificed three days later after radiation.

## Microbiota profiling

After the rats were sacrificed, midcolon contents were collected into several sterile EP (micro-centrifuge tube)tubes with corresponding numbers. The EP tubes were storaged in an ultra-low refrigerator at -80˚C and transported with dry ice. Genome-wide DNA was extracted from midcolon contents to amplify using specific primers with barcodes (16S V3 + V4). Paired end sequencing was performed on the Illumina MiSeq Platform and analyzed according to the previous study [19]. We used principal coordinate analysis (PCoA) to obtain principal coordinates from complex multidimensional data and visualize them. Diversity and inhomogeneity indicators including Shannon, Simpson, Chao1, and ACE were detected on this platform [20]. OTUs were further used by PICRUST for genome prediction of microbial communities [21]. Each group was subjected to three biological verifications.

## Gas chromatography-mass spectrometry

Preparation and High-Performance Liquid Chromatography (HPLC)-Based Analysis of FL on the Intestinal contents. 5g of mouse colon content was weighed and put into a beaker which was ground evenly after being dried in the shade, and 50mL methanol and a certain amount of sodium hydroxide was added and mixed until pH$> =$ 12. Then added 400mL water and 50mL dichloromethane in turn, vibrated for 5min and standed for 10min to separated the water phase and dichloromethane phase. The above solution was separated by the glass chromatographic column with 1g anhydrous sodium sulfate and 5g Florisil. Then added 20mL dichloromethane to elute the chromatographic column, elute twice, combine the eluent for three times, rotate and evaporate, concentrate to nearly dry, and then fix the volume to 1mL with n-hexane. HPLC analyses were performed on a Shimadzu liquid chromatography system (Shimadzu LC-2030, Kyoto, Japan) equipped with an evaporative light detector (Shimadzu 228–4511538, Kyoto, Japan). Chromatographic separation of the analytes was performed on a SinoChrom ODS-BP C18 analytical column (LC Column, 250 mm × 4.6 mm, 5 μm). The sample injection volume was 10 μL.

## Statistical analysis

The one-way analysis of variance (ANOVA) was used to analysis the statistical data in three or more groups. Levene's test was carried out to test the homogeneity of variances and followed with student's T test between the two groups (SPSS 20.0 software). The Mean±SEM was used to represent the data and denoted as follows: ns $P>0.05$; $^*P<0.05$; $^{**}P<0.01$; $^{***}P<0.001$.

# Result

## Imaging results of agarose gel electrophoresis

The extract DNA of each mid colon contents sample was collected by PCR agarose gel electrophoresis. 15000bp bands can be seen in each sample, indicating that every sample is qualified (Fig 1A). Agglomerate clustering is a hierarchical clustering algorithm. It is an unsupervised machine learning technique that partitions the population into several clusters, making data points more similar in the same cluster and different in different clusters. The result showed that the length of the branches of the same dose group was close and clustered together (Fig 1B).

## Radiation can reduce the number of OTUs

Operational Tax Units (OTUs) is often used in the analysis of microorganisms without culture, which can reverse the diversity and abundance of different microorganisms. Generally, if the

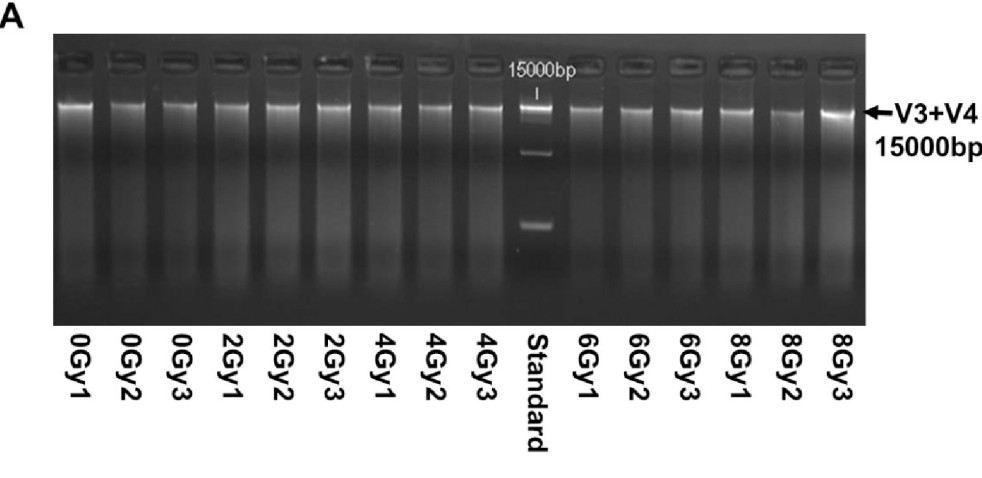

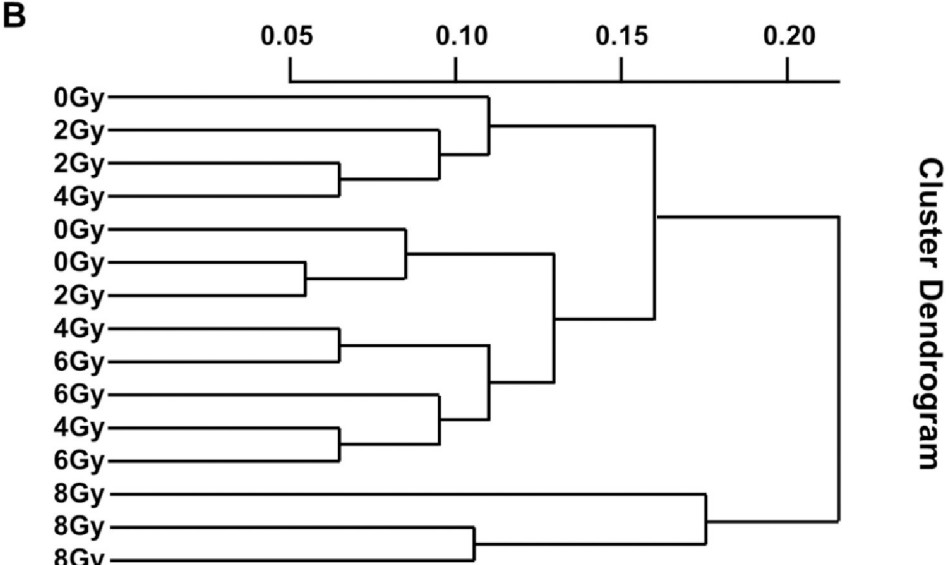

**Fig 1. Homogeneity detection of samples.** PCR agarose gel electrophoresis was used to detected of V3 and V4 protein content. We can see the 15000bp bands in each sample (A). The cluster analysis was provided by R software and the results of all samples are shown in the B.

similarity between sequences, such as different 16S rDNA sequences, is higher than 97%, it can be defined as an OTU. Fig 2A and 2B showed the result of the OTUs. The OTUs results showed that the total number of OTUs are 547 and they would decreased after irradiation in each group(Fig 2A).There are 254 similar OTUs in different groups (Fig 2B). There are 5 at 0Gy group different from the other groups, and there are 5 at 2Gy group, 3 at 4Gy group, 8 at 4Gy group, 3 at 8Gy group different from the other groups respectively (Fig 2B).

## Radiation can reduce the diversity of the intestinal flora

Alpha diversity (α diversity) refers to the diversity in a specific region or ecosystem, and the commonly used measurement indicators are as follows Shannon, Simpson, ace, Chao, etc. The greater the above index value, the higher the community diversity. Fig 3A–3D showed the result of the Alpha diversity. The results showed that with the increase of irradiation dose, the

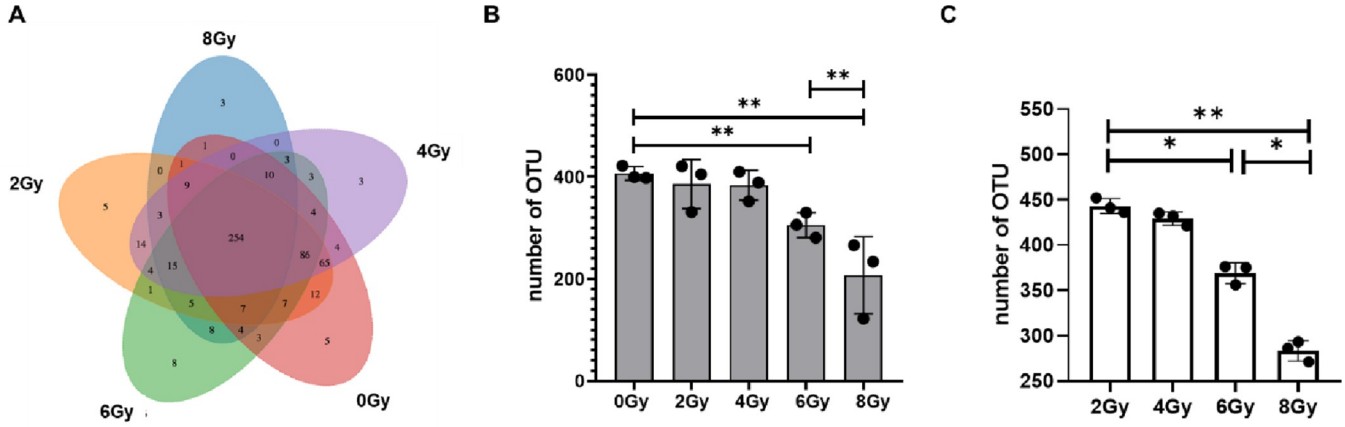

**Fig 2. Radiation can reduce the number of OTUs.** A shows the comparative Wayne diagram of OTUs composition. B shows the number of OTUs in each group after irradiation. Values are presented as mean±SEM, n = 3. Differences were assessed by ANOVA for multiple comparisons and denoted as follows: ns $P>0.05$; *$P<0.05$; **$P<0.01$; ***$P<0.001$.

α diversity of other groups decreased in a dose-dependent manner ($P<0.05$), and the α diversity of 6Gy and 8Gy groups decreased most obviously ($P<0.05$).

Beta diversity (β diversity) refers to the differences between samples, and the commonly used measurement indicators are as follows principal co-ordinates analysis (PCoA), Non-

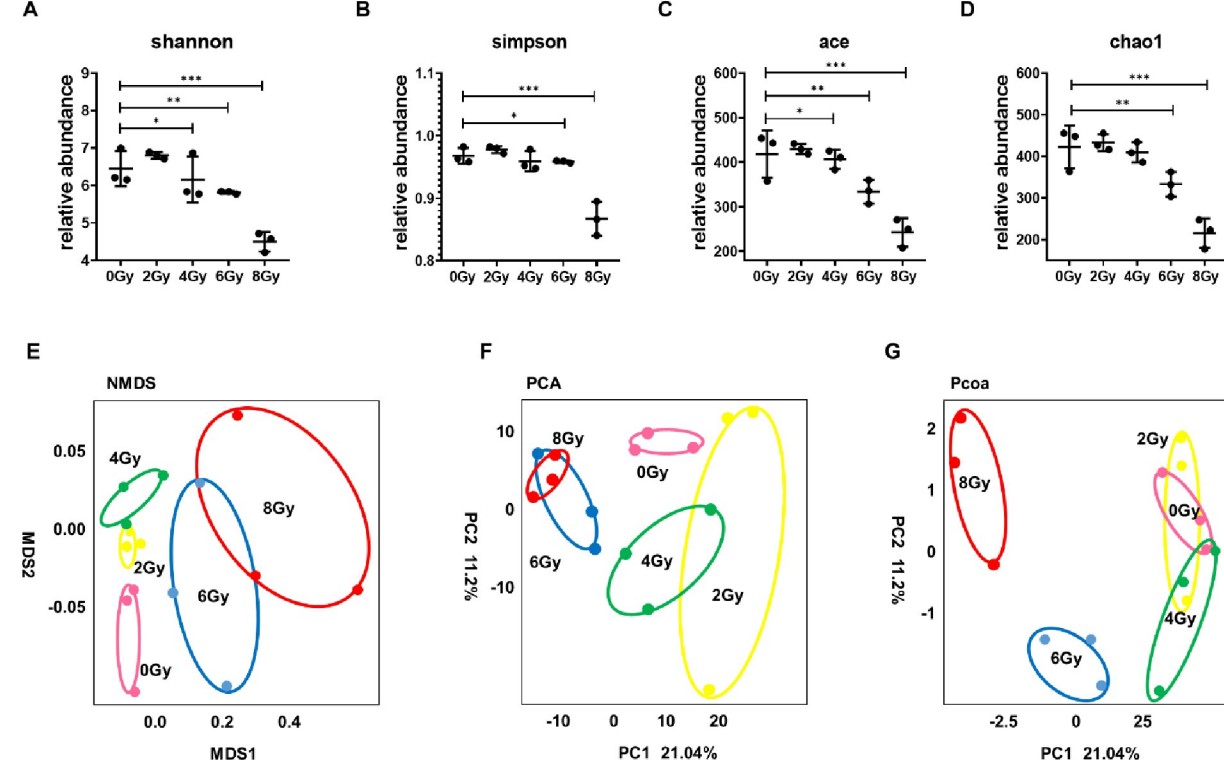

**Fig 3. Radiation can reduce the diversity of the intestinal flora.** A showed the result of the Shannon index in α-diversity analysis. B showed the result of the Simpson index in α-diversity analysis. C showed the result of the ace index in α-diversity analysis. D showed the result of the chao index in α-diversity analysis. E showed the result of the NMDS index in β-diversity analysis. F showed the result of the chao index in β-diversity analysis. G showed the result of the chao index in β-diversity analysis. Values are presented as mean±SEM, n = 3. Differences were assessed by ANOVA for multiple comparisons and denoted as follows: ns $P>0.05$; *$P<0.05$; **$P<0.01$; ***$P<0.001$.

metric Multidimensional Scaling (NMDS) and Principal component analysis (PCA). PCoA is a visualization method to study the similarity or difference of data, which can be used to observe the differences between individuals or groups. NMDS is often used to compare the differences between sample groups, which can be based on evolutionary relationship or quantitative distance matrix. PCA is a visual method to study the similarity or difference of data and it can find the most important coordinates in the distance matrix. Each point in the graph represents a sample, and the points with the same color come from the same group. The closer the distance between two points, the smaller the difference in community composition between them. Fig 3E–3G showed the result of the β diversity. The results show that the distance between samples in 0Gy, 2Gy and 4Gy groups is close and difficult to distinguish, while samples in 6Gy and 8Gy occupy their own space and are easy to distinguish.

## Radiation can change overall microbial composition in each group

The overall microbial composition in each group differed at the phylum and genus levels (Fig 4A–4N). The largest phylum represented in each dataset was *Firmicutes*, *Bacteroidetes*, and *Proteobacteria* (Fig 4A, 4C–4E). The abundance level of *Firmicutes* enhanced when irradiated by 2Gy, but did not change obviously when irradiated by 4Gy and 6Gy ($P>0.05$), and attenuated when irradiated by 8Gy ($P<0.05$). The abundance level of *Bacteroidetes* decreased when irradiated by 2Gy, increased when irradiated by 4Gy and 6Gy, and attenuated when irradiated by 8Gy ($P<0.05$). The abundance ratio of *Firmicutes/Bacteroides* is inverted. The abundance level of *Proteobacteria* did not change obviously when irradiated by 2Gy, 4Gy and 6Gy ($P>0.05$), but increased significantly when irradiated by 8Gy ($P<0.05$).

At the genus level, the largest genus represented in each dataset was *Phascolarctobacterium*, *Ruminococcaceae*, *Christenellaceae*, *Lactobacillus*, *Prevotellaceae*, *Lachnospiraceae*, *Bacteroides*, *Lachnoclostridium* and *Escherichia-Shigella* (Fig 4B, 4F–4N). The abundance level of *Phascolarctobacterium Ruminococcaceae* and *Lachnospiraceae* increased at 2Gy, and decreased at 4Gy, 6Gy and 8Gy($P<0.05$). The abundance level of *Christenellaceae* and *Lactobacillus* decreased in a dose-dependent manner, while the abundance level of *Lachnoclostridium* increased in a dose-dependent manner. The abundance level of *Prevotellaceae* decreased at 2Gy, and increased at 4Gy, 6Gy and 8Gy ($P<0.05$). The abundance level of Bacteroides didn't change obviously at 2Gy ($P>0.05$), but increased significantly at 4Gy, 6Gy and 8Gy($P<0.05$). The abundance level of *Escherichia-Shigella* didn't change obviously at 2Gy, 4Gy and 6Gy ($P>0.05$), but increased significantly at 8Gy ($P<0.05$).

## Biomarkers of intestinal flora at different irradiation doses

Lefse is based on linear discriminant analysis, which combines linear discriminant analysis with nonparametric Kruskal-Wallis and Wilcoxon rank sum test to screen Biomarker between groups (Fig 5A and 5B). It can be seen from the figure that the biomarker of 0Gy group is *Veillonellaceae*, the biomarker of 2Gy group is *Firmicutes*, the biomarker of 4Gy group is *Dehalobacterium* and *Dehalobacteriaceae*, and the biomarker of 6Gy group is *Odoribacter*. The biomarkers of 8Gy group are *Anaerotruncus*, *Holdemania*, *Proteus*, *Bilophila*, *Desufovibrionales* and *Deltaproteobacteria*.

## Radiation can decrease the expression of the SCFAs in colon contents

Short chain fatty acids include Acetic acid, propionic acid, and butyric acid mainly, which were detected by gas chromatography-mass spectrometry. The abundance level of Acetic acid

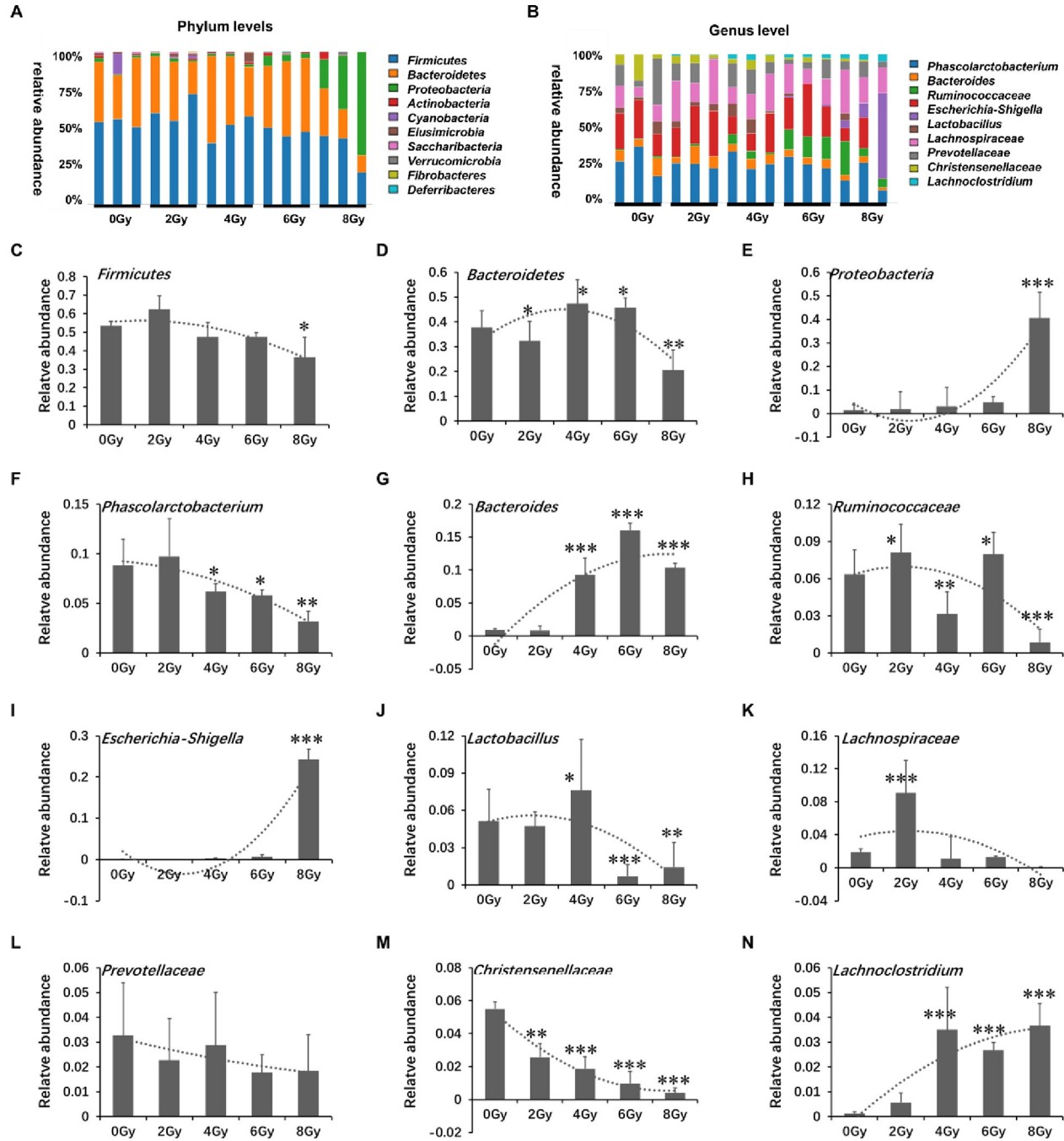

**Fig 4. Radiation can change overall microbial composition in each group by X ray.** Microbiota compositions at the phylum level (A) and at the genus level (B) have been shown in this figure. The figures show bacteria with obvious changes at the phylum level (C-E). The figures show bacteria with obvious changes at the genus level (F-N). Values are presented as mean±SEM. Differences were assessed by ANOVA for multiple comparisons and denoted as follows: ns $P>0.05$; *$P<0.05$; **$P<0.01$; ***$P<0.001$.

decreased in a dose-dependent manner, while the abundance level of propionic acid, and butyric acid decreased sharply when at 2Gy, 4Gy, 6Gy and 8Gy (Fig 6A–6C). So radiation markedly reduced Acetic acid, propionic acid, and butyric acid.

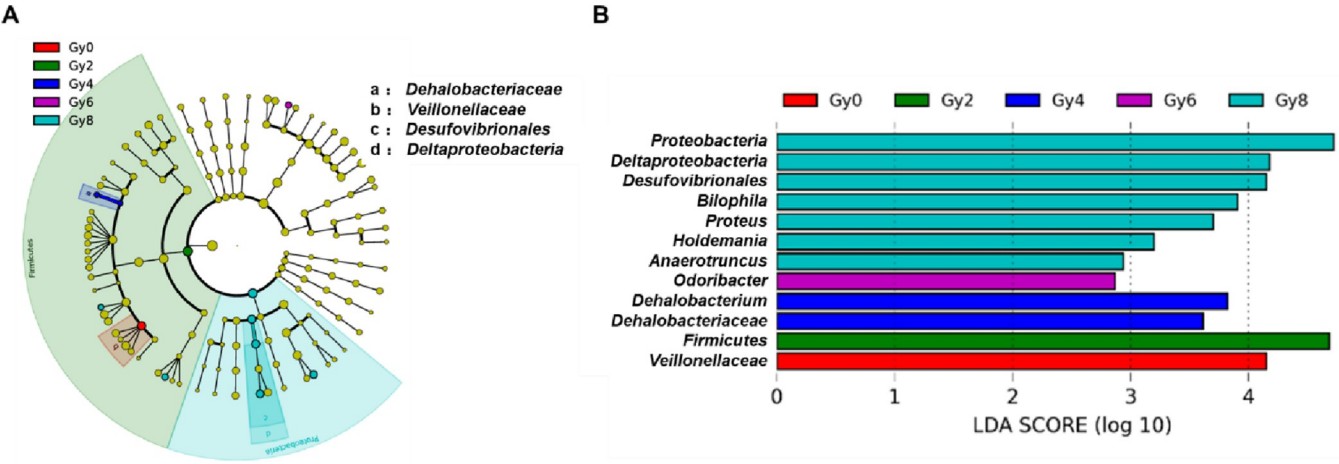

**Fig 5. The analysis of LDA difference contribution interpretation of the map.** Different colors in the map represent significantly different species between different samples or groups. It is obtained by analysis with LefSe software, in which the significant difference of the logarithmic LDA score is set to 2.

## Discussion

Radiotherapy is a common method for the treatment of malignant tumors, and X-ray is one of the most frequently used measure. However, it inevitably damages the normal tissues and intestinal homeostasis [22]. Intestinal symbiotic bacteria are a kind of bacteria that can resist the colonization of foreign microorganisms. They act together on the body to play a total defense function, which is called biological barrier. When the intestinal microecological homeostasis is out of balance, the colonization resistance of the intestinal tract to external pathogens decreases, which eventually leads to the invasion and pathogenicity of external bacteria, and even the potentially pathogenic bacteria in the intestinal tract begin to play their pathogenic role. The intestinal tract has the largest "reservoir" in the body. It can produce a large amount of endotoxin and enhance the damage induced by endotoxemia, significantly increasing the mortality of acute and critical diseases [23]. Previous studies have shown that radiation is one of the factors causing intestinal microecology imbalance [24]. In this experiment, we selected different doses of X-ray to radiate SD rats to evaluate the effect of different doses of X-ray on intestinal flora imbalance.

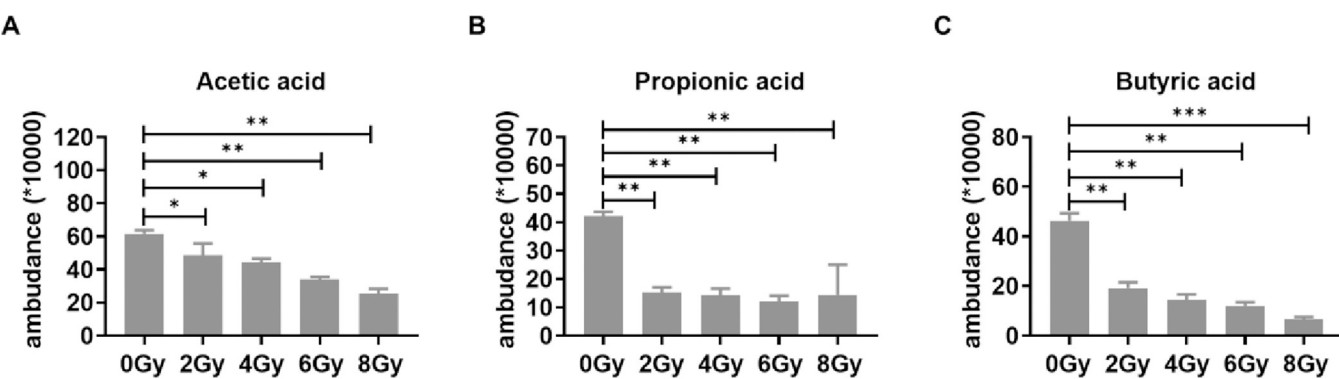

**Fig 6. Radiation can decrease the expression of the SCFAs in colon contents.** Acetic acid (A), propionic acid (B), and butyric acid (C) were detected by gas chromatography-mass spectrometry. Values are presented as mean±SEM. Differences were assessed by ANOVA for multiple comparisons and denoted as follows: ns $P>0.05$; *$P<0.05$; **$P<0.01$; ***$P<0.001$.

In this study, 15000bp bands can be seen in each sample (Fig 1A) and the length of the branches of the same dose group was close and clustered together (Fig 1B), indicating that every sample is qualified. The OTUs results showed that the number of OTUs decreased after irradiation at 6Gy and 8Gy (Fig 2). Shannon, simpson, ace and chao1 indexes were examined for the alpha-diversity of the microbiomes. The above indexes were increased at 2Gy, decreased at 4Gy, 6Gy and 8Gy compared to controls (Fig 3A–3D). To assess overall differences in beta diversity, we applied NMDS, PCoA maps, and Pca spatial maps to weighted and unweighted UniFrac distance metric matrices generated for the sample set. Control and irradiated rats showed significant microbial community structure aggregation (Fig 3E–3G). This result further indicates that the type and abundance of intestinal microorganisms in rats can be changed from 6Gy total body irradiation dose, which is widely involved in various physiological activities of the body and is called "metagenomic" (Also called microbial environmental genomics, metagenomics. The metagenomic library is constructed by directly extracting the DNA of all microorganisms from environmental samples, and the genetic composition and community function of all microorganisms in environmental samples are studied by using the research strategy of genomics) intestinal flora [17], and appears disorder at 6Gy and 8Gy irradiation dose. The intestinal flora response of low-dose and high-dose irradiation was not exactly the same. However, there is evidence showed that a higher richness and diversity in *B. dorsalis* flies [25] and a lower diversity in rats [16] and patients [13]. This is not exactly the same as our study, the reason maybe that different species have different responses to radiation.

As several previous studies on the mammalian gut have included human and ratio [26], *Bacteria phyla* and *Firmicutes* were identified as two major phyla in the small and large intestines of mice. Although there were differences in the abundance of multiple phyla between control and different dose irradiated groups, the abundance ratio of *Firmicutes/Bacteroides* was opposite to that of control. Notably, *Proteobacteria* was identified in the irradiated large intestines at 8Gy, control samples and other dose samples, however, did not, suggesting that proliferation of microbial cells belonging to this phylum may increase after irradiation. The study have shown that *Proteobacteria* is the key microorganism to induce enteritis, which may be one of the markers of disease [27]. That means high doses radiation can induce the growth of pernicious bacteria.

At the genus level, 16sr RNA analysis showed that the abundance levels of *Phascolarctobacterium*, *Ruminococceae* and *Lachnospiraceae* decreased significantly at 4Gy, 6Gy and 8Gy. *Phascolarctobacterium* produce short chain fatty acids, which play a variety of important roles in maintaining human health, such as enhancing gastrointestinal function and reducing the level of inflammation [28]. The main metabolites of *Ruminococcaceae* are acetic acid and formic acid [29]. The bacteria in *Ruminococcaceae* mainly rely on absorption of monosaccharide and degradation of mucin to obtain energy. *Lachnospiraceae* may be a potential beneficial bacterium, which is involved in the metabolism of many carbohydrates [30]. The fermentation product acetic acid is the main source of energy for the host. The main metabolites of above three are Acetic acid propionic acid and Valeric acid, giving a hint that the decrease of short chain fatty acids may be related to the decrease of bacteria producing short chain fatty acids. *Lactobacillus* and *Christenellaceae* were considered as beneficial agents who are benevolent workers that help digest food [31], however the abundance levels of them radiation decreased in a dose-dependent manner compared with control group. Studies have shown that *Lactobacillus* is beneficial to the health of the host, which can ferment sugars to produce lactic acid, maintain human health and regulate immune function [32]. *Christenellaceae* is beneficial to human health, and its abundance is higher in thinner people [33]. The abundance of *Prevotellaceae* increased with the irradiation of 4Gy, 6 Gy and 8Gy. Studies have shown that

*Prevotellaceae* is one of the common bacteria causing endogenous infection, and the pathogenic substance may be collagenase, which is related to the decomposition of connective tissue. *Lachnoclostridium* is tend to exist in obese patients, it ascends in a dose-dependent manner after radiation [34]. Bacteroides are symbiotic bacteria in human intestines, which provide essential nutrients for the body [35]. However, when they enter other parts of the body other than the gastrointestinal tract, they will lead to the aggravation of abscess and other infections, and the abundance level will increase significantly at 4Gy, 6Gy and 8Gy irradiation. *Escherichia-Shigella* has been proved is the pathogen of dysentery and it is also one of the pathogenic bacteria that enter the blood after the destruction of intestinal barrier [36]. In this report, the abundance of Escherichia-Shigella was only dramatically elevated by radiation at 8Gy group. These three kinds of pathogenic bacteria showed a significant increase in different groups, unexpectedly the degree of their increase was different.

We have also identified some potential markers that may play a therapeutic role in different treatments. It was an interesting finding that the biomarkers of 2Gy group were *Firmicutes*, 4Gy group were *Dehalobacterium* and *Dehalobacteriaceae*, 6 Gy group were *Odoribacter*, 8Gy group were *Anaerotruncus*, *Holdemania*, *Proteus*, *Bilophila*, *Desufovibrionales* and *Deltaproteobacteria*. This suggests that moxibustion and electroacupuncture may have some similar therapeutic targets. Although we have found some potential biomarkers, how to regulate these markers by moxibustion and electroacupuncture still requires further research. These findings indicate that the microbiota emerges as a critical driver of radiation-response, but still need to pose several questions. (1) There are only three samples in each group, so the sample size is too small. (2) how do specific bacterial species modulate inflammatory responses? (3) Whether these findings could translate into a therapeutic approach for the treatment of radiation-induced diseases in humans deserves further attention.

## Supporting information

**S1 Raw images.**
(RAR)

## Author Contributions

**Conceptualization:** Liying Zhang, Yongqi Liu.

**Data curation:** Yangyang Li.

**Formal analysis:** Ting Zhou.

**Investigation:** Xiaomin Xu.

**Project administration:** Yiming Zhang.

**Writing – review & editing:** Zhiming Miao.

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
