## [Decision Letter · Decision Letter 0]

27 Mar 2023

PONE-D-22-28249A potential marker of radiation based on 16S rDNA in the rat model: intestinal floraPLOS ONE

Dear Dr. Yongqi Liu,

Thank you for submitting your manuscript to PLOS ONE. After careful consideration, we feel that it has merit but does not fully meet PLOS ONE’s publication criteria as it currently stands. Therefore, we invite you to submit a revised version of the manuscript that addresses the points raised during the review process.

We look forward to receiving your revised manuscript.

Kind regards,

Awatif Abid Al-Judaibi, PhD

Academic Editor

PLOS ONE

- https://pubmed.ncbi.nlm.nih.gov/25600706/#:~:text=The%20gut%20microbiotas%20in%20control,intestines%20at%20the%20genus%20level.

- https://www.hindawi.com/journals/ecam/2020/9017854/

In your revision ensure you cite all your sources (including your own works), and quote or rephrase any duplicated text outside the methods section. Further consideration is dependent on these concerns being addressed.

5. Thank you for stating the following in the Acknowledgments/ Funding Section of your manuscript:

“This study was funded by Natural Science Foundation of Gansu Province (NO. 20JR10RA332, 20JR10RA318). National Natural Science Foundation of China (NO. 82260882, 82004094). China Postdoctoral Science Foundation Project (NO. 2021M693794). Lanzhou City Health Key Science and Technology Development Project (NO. 2021006).”

“NO - Include this sentence at the end of your statement: The funders had no role in study design, data collection and analysis, decision to publish, or preparation of the manuscript.”

6. Thank you for stating the following in your Competing Interests section: 

“NO authors have competing interests”

7. PLOS ONE now requires that authors provide the original uncropped and unadjusted images underlying all blot or gel results reported in a submission’s figures or Supporting Information files. This policy and the journal’s other requirements for blot/gel reporting and figure preparation are described in detail at https://journals.plos.org/plosone/s/figures#loc-blot-and-gel-reporting-requirements and https://journals.plos.org/plosone/s/figures#loc-preparing-figures-from-image-files. When you submit your revised manuscript, please ensure that your figures adhere fully to these guidelines and provide the original underlying images for all blot or gel data reported in your submission. See the following link for instructions on providing the original image data: https://journals.plos.org/plosone/s/figures#loc-original-images-for-blots-and-gels.

Reviewers' comments:

Reviewer's Responses to Questions

**Comments to the Author**

1. Is the manuscript technically sound, and do the data support the conclusions?

Reviewer #1: Yes

Reviewer #2: Yes

2. Has the statistical analysis been performed appropriately and rigorously? 

Reviewer #1: Yes

Reviewer #2: Yes

3. Have the authors made all data underlying the findings in their manuscript fully available?

Reviewer #1: Yes

Reviewer #2: Yes

4. Is the manuscript presented in an intelligible fashion and written in standard English?

Reviewer #1: Yes

Reviewer #2: No

5. Review Comments to the Author

Reviewer #1: Reviewer comments are attached in the manuscript pdf file. overall, authors performed a rigorous technic and methodology. A a further explanation and discussion are required. Additional figure to describ the study design is needed.

Reviewer #2: Recent study in microbiome is just providing insight into the composition, pattern, and predict functions of the intestinal microbiome in SD rats, and certain abundant genera and species could serve as a biomarker for diagnosis and treatment. in this regard this MS is timely and valuble.

1. Requires edition> Example: Abstract part (line 5. Microbiotas. Punctuation: line. 28 change .). Ethics statement: line 5. 222℃. EP tubes. it is not usual, to begin with, a number (5g of a mouse). (Five grams of…). SD rats. In its first use: Sprague Dawley rates.

2. Introduction. After irradiation, it can cause DNA double-strand or single-strand breaks and other biological effects. It has been shown that the gut microbiota regulates the efficacy of radiotherapy and chemotherapy through a "TIMER" mechanism, which suggests a reduction in translocation, immune regulation, metabolism, enzymatic degradation, and diversity. Requires citation.

. Casero found … David C et al in 2017 found...

. Zhang also shows…. Zhang A and Steen TY in 2017 also showed...

3. They act together on the body to play a total defense function, which is called a biological barrier. .... begin to play their pathogenic role. Citation Required.

4. Result From 16S rDNA sequencing: How many counts were generated and what % of proportion clustered with similarity indicated unique OTUs?

5. Your mention of the “metagenomic" intestinal flora is not clear for readers and requires further clear writing. As several previous studies on the mammalian gut have included humans and ratio [27] clarify.

6. These could be the limitations of the study

(1) There are only three samples in each group, so the sample size is too small. (2) how do specific bacterial species modulate inflammatory responses?

7. This could be your recommendation: (3) Whether these findings could translate into a therapeutic approach for the treatment of radiation-induced diseases in humans deserves further attention.

6. PLOS authors have the option to publish the peer review history of their article (what does this mean?). If published, this will include your full peer review and any attached files.

Reviewer #1: **Yes: **Arif Sabta Aji

Reviewer #2: **Yes: **Gizachew Taddesse AKALU

---

## [Author Response · Author response to Decision Letter 0]

20 Apr 2023

Reviewer #1: Reviewer comments are attached in the manuscript pdf file. overall, authors performed a rigorous technic and methodology. A a further explanation and discussion are required. Additional figure to describ the study design is needed.

Due to the need to upload the reply to the reviewer separately, we have sorted out and answered the opinions of the reviewer below.

1.In the method section, please add the statistical method information that had been used in the study in concisely.

Hello, reviewer, thank you for your comments. In the text, we introduce the statistical analysis. In order to prevent excessive abstract word count, the statistical analysis was not described in the abstract.

2.Please add one sentence for recommendation based on the findings of the study.

Hello, reviewer, thank you for your comments, which have been revised in the original text.

3.Grouping of SD rats should be shown in figure. 

Hello reviewer, thank you for your opinion. The grouping here is consistent with the grouping in the following text.

4.What is EP stand for?

Hello reviewer, EP tubes means a microcentrifuge tube.

5.where this activity was performed?

Dear reviewer, thank you for your question. We operated in the professional animal materials room of Gansu University of Traditional Chinese Medicine.

6.where the conclusion section is?

Hello reviewer, thank you for your question. Since our paper contains many results, we integrated the conclusion with the discussion. Written in the discussion section.

Reviewer #2: Recent study in microbiome is just providing insight into the composition, pattern, and predict functions of the intestinal microbiome in SD rats, and certain abundant genera and species could serve as a biomarker for diagnosis and treatment. in this regard this MS is timely and valuble.

1. Requires edition> Example: Abstract part (line 5. Microbiotas. Punctuation: line. 28 change.). Ethics statement: line 5. 222℃. EP tubes. it is not usual, to begin with, a number (5g of a mouse). (Five grams of…). SD rats. In its first use: Sprague Dawley rates.

Hello, reviewer, thank you for your comments, which have been revised in the original text.

2. Introduction. After irradiation, it can cause DNA double-strand or single-strand breaks and other biological effects. It has been shown that the gut microbiota regulates the efficacy of radiotherapy and chemotherapy through a "TIMER" mechanism, which suggests a reduction in translocation, immune regulation, metabolism, enzymatic degradation, and diversity. Requires citation.

. Casero found … David C et al in 2017 found...

. Zhang also shows…. Zhang A and Steen TY in 2017 also showed...

Hello, reviewer, thank you for your comments, which have been revised in the original text.

3.They act together on the body to play a total defense function, which is called a biological barrier. .... begin to play their pathogenic role. Citation Required.

Hello, reviewer, thank you for your comments, which have been revised in the original text.

4.Result From 16S rDNA sequencing: How many counts were generated and what % of proportion clustered with similarity indicated unique OTUs?

Hello reviewer, our calculated number is 547.Such as different 16S rDNA sequences, is higher than 97%, it can be defined as an OTU.

5.Your mention of the “metagenomic" intestinal flora is not clear for readers and requires further clear writing. As several previous studies on the mammalian gut have included humans and ratio [27] clarify.

Hello, reviewer, thank you for your comments, which have been revised in the original text.

6. These could be the limitations of the study

(1) There are only three samples in each group, so the sample size is too small. (2) how do specific bacterial species modulate inflammatory responses?

Hello reviewer, thank you for your comments, (1) we wrote 3 for the 3 animals, but we had 9 mice in each group during the experiment.(2) The content of this part will be explored in the following experiment. Thank you for your advice, which is very helpful to us.

7. This could be your recommendation: (3) Whether these findings could translate into a therapeutic approach for the treatment of radiation-induced diseases in humans deserves further attention.

Hello, reviewer, thank you for your suggestions. We are also thinking about whether suitable natural drugs can be found to target the results of this paper. Your advice is very helpful to us, and we will think about it seriously and explore it in the following work.

---

## [Editor Report · Decision Letter 1]

7 May 2023

A potential marker of radiation based on 16S rDNA in the rat model: intestinal flora

PONE-D-22-28249R1

Dear Dr. Yongqi Liu,

We’re pleased to inform you that your manuscript has been judged scientifically suitable for publication and will be formally accepted for publication once it meets all outstanding technical requirements.

Kind regards,

Awatif Abid Al-Judaibi, PhD

Academic Editor

PLOS ONE

Additional Editor Comments (optional):

Dear Authors,

Thank you for your positive response to the reviewers' comments, I submitted final decision as accept, but two comments I requested you to revised them in the final document:

1. comment number 4 of reviewer 1, you answer the question but did not add the abbreviation meaning in the article. Kindly add them to the article, this will help readers to follow your methodology.

2. comment number 4 of reviewer 2, you added the percentage but not the counted number. Also, it will be helpful to readers if you add the counted number.

---

## [Editor Report · Acceptance letter]

20 Jul 2023

PONE-D-22-28249R1 

A potential marker of radiation based on 16S rDNA in the rat model: intestinal flora 

Dear Dr. Liu:

I'm pleased to inform you that your manuscript has been deemed suitable for publication in PLOS ONE. Congratulations! Your manuscript is now with our production department. 

Kind regards, 

on behalf of

Professor Awatif Abid Al-Judaibi 

Academic Editor

PLOS ONE